# Intersectionality of the Gender Wage Gap Among Healthcare Professionals: A Scoping Review

**DOI:** 10.3390/healthcare13030273

**Published:** 2025-01-30

**Authors:** Neeru Gupta, Jonathan Zoungrana

**Affiliations:** Department of Sociology, University of New Brunswick, Fredericton, NB E3B 5A3, Canada

**Keywords:** scoping review, gender equality, gender wage gap, social discrimination, healthcare professionals, physician remuneration

## Abstract

***Background:*** A growing body of research has documented persistent wage gaps between women and men in the healthcare workforce, a pattern widely observed across cadres and countries. Less well known is whether various intersecting characteristics often associated with social discrimination may exacerbate or attenuate gendered disparities. This review scopes contemporary research from diverse settings focusing on how race, ethnicity, and sexual and gender minority status may intersect in shaping earnings differentials among healthcare practitioners to help inform policy and management decisions. ***Methods:*** Studies quantifying the intersecting axes of gender and other postulated social drivers of differed practitioner earnings were identified by systematically searching five bibliographic databases (Embase, CINAHL, EconLit, SocIndex, and PsychInfo) and scanning the reference lists of review articles and other forms of the global health literature. A total of 2123 reports were retrieved; after screening, 21 articles were retained for narrative synthesis. ***Results:*** The studies covered data from four countries (Brazil, Norway, the United Kingdom, and the United States). Physicians were researched most often (43% of the synthesized articles) followed by nurses (38%). No uniform patterns were found in gendered earnings variations stratified by race, ethnicity, and/or ancestry; however, wide variations were seen in the way the relationships were operationalized across studies and contexts. One investigation included sexual orientation as a factor in earnings gaps, but presented results combined with other personal characteristics. None of the studies examined wage data by gender minority status. ***Conclusions:*** This review highlighted notable limitations in the available research in relation to disaggregated measures of ethnocultural heterogeneity, robust methodologies and transparent reporting, and the underlying health workforce information systems for incorporating more diversity elements and enhancing cross-national comparability in assessments of structural wage gaps among healthcare practitioners.

## 1. Introduction

A growing body of the literature on human resources in healthcare systems documents clear gender-based pay gaps, with women’s earnings averaging significantly less than men’s, a pattern widely observed within and across countries [1,2,3,4]. Given that women account for the strong majority (over two-thirds) of the health and social care workforce worldwide, such underinvestment in health practitioners undermines the prospects of achieving the Sustainable Development Goals and universal health coverage [3]. Boniol and colleagues assessed an unexplained gender pay gap of 11% in the global health workforce, after accounting for differences in working hours and in occupational trajectories, e.g., in traditionally male-dominated medicine, dentistry, and pharmacy versus female-dominated nursing and midwifery [3]. Several health professional associations—including but not limited to the World Medical Association, International Council of Nurses, and World Physiotherapy—have recognized the need to redress inequities in pay and other health labour market challenges negatively affecting female practitioners, particularly given women’s uniquely imbalanced childbearing and childrearing roles combined with social biases favouring leadership opportunities and other workplace rewards among men [5,6,7]. Where the global evidence base has been less apparent is the various intersecting social drivers of such persistent female/male earnings disparities, such as reinforcing effects by race, ethnicity, sexual orientation, gender diversity, and other characteristics often associated with social discrimination and differential treatment in the workplace. Attention to the need for policy-actionable data and research on multiple forms of structural inequalities has risen in recent years, accelerated by the COVID-19 pandemic and the resultant health workforce crisis [6,8].

Many workplace differentials in economic rewards observed across sectors (public, private, and multi-sector environments) and around the globe cannot be explained by human capital and skill competencies alone, but rather by the effects of non-work-related factors which may induce pay gaps, such as gender and social discrimination [2]. Unequal pay for work of comparable value can lead to health worker turnover and burnout, negatively impacting quality of care and health system performance [1]. At the organizational level, two main theories have been postulated in the labour economics literature to account for unexplained differences in professional earnings across groups of workers (e.g., men versus women, majority versus minority ethnic groups): statistical discrimination and taste-based discrimination [9,10]. Taste-based discrimination assumes that individual prejudice among employers, colleagues, and clients influences hiring, promotion, and valuation decisions; it is presumed to decrease with increasing social and socioeconomic equity and inclusion [9]. Statistical discrimination applies the notion of attributing ambiguous quantifiable information about workers’ wages in relation to productivity-related characteristics (e.g., education, language competence) to unmeasured inequalities in labour market conditions. Much of the existing empirical literature on wage gaps overall focuses on statistical discrimination approaches [9].

Understanding the social interactions attenuating or exacerbating gender gaps in earnings as a manifestation of health labour market discrimination is prerequisite to developing appropriate policies and management practises to advance diversity, equity, inclusion, and social justice. Despite the widespread evidence of differential treatment in the health workforce, disentangling the drivers of earnings gaps against what would be predicted statistically in truly ‘gender-blind’ and ‘colour-blind’ societies is challenging; however, it has been argued that the more employers capture information about their workers, the more likely salaries may be dependent on measures of productivity and less so on observable personal characteristics [10]. In this review, we examine the state of the literature quantifying different dimensions of social minority status with experiences of devaluation of women’s labour in the healthcare workforce. Given the aim to characterize the nature and extent of research activity and to identify recurring themes on this issue, we use a scoping review approach [11,12,13]. We focus on peer-reviewed research published since the year 2000, aligned with the growing interest in how pay gaps have evolved in more recent times and given the recognition that strategies for tackling health workforce shortages require consideration of gender and ethnocultural diversification of practitioners. Following this introduction, this review is structured according to the framework for scoping studies by Arksey and O’Malley [13]: identification of the research question; identification and selection of relevant studies; charting the data; and collating, summarizing, and reporting the results.

## 2. Methods

### 2.1. Research Question

The research question of interest was as follows: How does social minority status (based on race, ethnicity, sexual orientation, and/or gender-diverse identity) intersect with observed wage disparities between women and men healthcare practitioners? To address this question, we focused on primary quantitative studies of gender wage gaps underlined by taste-based or statistical discrimination in the health labour market. We conducted our review based on a PICOS (Population, Intervention, Comparison, Outcomes, Study type) framework, and in line with the *Preferred Reporting Items for Systematic Reviews and Meta-analyses* (PRISMA) guidelines [14].

### 2.2. Identification of Relevant Sources and Study Selection

We systematically searched for relevant peer-reviewed publications in five specialized (biomedical and non-biomedical) abstract and citation databases: Embase, CINAHL, EconLit, SocIndex, and PsychInfo. The search strategy was developed with the advice and assistance of a library professional. Free text and formal search terms and filters reflecting the question at hand were translated to respect database-specific requirements. Several medical subject heading (MeSH) terms and combinations were used to identify the health workforce population [including “physician*”, “nurse*”, “dentist*”, “physiotherapist*”, “pharmacist*”, and related nomenclature] and the outcome of interest [including “earning*”, “income inequality”, “pay gap”, “labour discrimination” and related nomenclature]. Eligible to be included in the review were original research studies that addressed the question through some sort of quantitative evaluation component, published between 2000 and 2023, in English or French. No restrictions were placed by world region, country income level, healthcare financing system, health worker cadre, or professional regulatory status. Although health worker remuneration can comprise different types of payments, we followed the common practice of using the terms earnings, incomes, wages, and salaries interchangeably. A complete illustrative search strategy for one of the databases can be found in Appendix A (Table A1).

To find additional potentially eligible studies, we manually reviewed the reference lists of previous systematic reviews on gender pay gaps [1,2,4] as well as recent international reports from the World Health Organization (WHO) on pay gaps and other gendered assessments of the health workforce [15,16,17]. The WHO acknowledges a broad range of societal drivers may intersect with gender in influencing and reinforcing discrimination and inequities in health labour markets—including race/ethnicity, sexual orientation, and other stratifiers of access to opportunities, remuneration, and power in decision-making [16]. We did not consider migrant pay gaps, that is, studies collapsing all foreign-born or foreign-trained workers together, without distinction for race/ethnicity/nationality, given that these may be related more closely to changes in admission policies and other politically driven factors [10]. Also excluded were studies focusing exclusively on unpaid caregivers or non-pecuniary employment benefits. In accordance with our review protocol (not published), a description of the inclusion and exclusion criteria is found in Table 1.

### 2.3. Data Charting Process and Synthesis of Results

To secure consensus on studies for review inclusion, two reviewers independently screened a sample of the retrieved abstracts, and in turn all retrieved full-text articles. We recorded the study objective, the country/study setting, the healthcare worker and/or patient population in the study, the data gathering technique, the comparison groups, and the outcome measured. Each study’s contents were vetted by three items based on the level on inclusion and reporting of intersectional gender data and analysis [18], notably whether they (i) explicitly defined and distinguished “sex” (i.e., as a biological variable) from “gender” (as a social construct influenced by cultural and behavioural norms and self-identity); (ii) explicitly defined and analyzed another aspect of social identity often compared in relation to a dominant group in a culture (with focus here on race, ethnicity, or ancestry; sexual minority status; gender minority status); and (iii) applied quantitative methods to examine the combined effects on professional earnings of gender and at least one other intersecting social identity factor (e.g., using statistical interaction or decomposition methods on population-level microdata).

We conducted a narrative synthesis to interpret the data. Because of the heterogeneity typical in pay gap research of target populations, outcome measurements, confounding factors examined, and research protocols and norms [2], we did not expect to be able to perform a meta-analysis.

## 3. Results

### 3.1. Selection and Characteristics of the Studies

We screened 2123 records in total: 2108 records from the five electronic databases, along with 15 records from hand-searching through systematic reviews and grey literature sources. In a first step, we removed 2086 duplicates and other records based on title and abstract screening. We screened for the type and language of publication, target population (i.e., category of healthcare providers), study design (distinguishing sex/gender plus another intersecting social characteristic), and outcome of interest (professional earnings). Following this initial screening, we retained 37 articles for full-text review, of which 16 were eventually screened from further consideration, leaving 21 studies retained for narrative synthesis. A PRISMA depiction of the flow of information is charted in Figure 1.

Following the PICOS framework, characteristics of the 21 studies are described in Table 2 [19,20,21,22,23,24,25,26,27,28,29,30,31,32,33,34,35,36,37,38,39]. The majority (81%) of the studies focused on the United States, either alone or in a cross-national perspective. Two studies focused on the United Kingdom, and one each on Brazil, Canada, and Norway.

The most covered professions were physicians (43% of the studies) and nurses (38%), either alone or in a cross-professional perspective. Additional covered professions included dentists, pharmacists, physician assistants, and other non-physician service providers (e.g., psychologists, social workers). Study designs ranged from ad hoc surveys of relatively small samples and based on specific selection criteria with variable response rates (e.g., [20,24,26]), to large-scale population-based enumerations from census or registry data (e.g., [25,29,36,39]). As expected, remunerations were measured heterogeneously, such as employer-assigned salaries per job grade, wages from the primary work position through self-reports, or total annual earnings from all positions through tax records.

All of the synthesized studies included data on race, ethnicity, and/or ancestry of the person as a purported social characteristic influencing wage differentials. One study further investigated sexual orientation as an explanatory variable. None included measures of gender minority (e.g., transgender, non-binary) identities.

### 3.2. Measurements of Intersecting Social Characteristics

Across the five country contexts captured in the review, differences were evident in the unique ways social identities were conceptualized and measured in relation to race, ethnicity, and ancestry. In the United States, studies entailing secondary analyses of population-based survey data delineated ‘race’ by the standard categories of the US Census Bureau (e.g., [23,25]). In this context, federal statistical standards emphasize using self-identification as the “preferred means of obtaining information about an individual’s race and ethnicity” and discourage designating any given population groups as “minority groups”; the standard also differentiates questions on race (e.g., White, Black or African American, American Indian) from “Hispanic or Latino” origin (e.g., Puerto Ricans) [40]. Reviewed studies using primary data collection tended to follow the same groupings. Some researchers specifically focused on differences between White and Black practitioners (e.g., [21,28,38]), given the posited persistence of “racially embedded processes of work” stemming from the country’s history of legal racial segregation [28]. Others framed labour inequalities among practitioners as salient to wider social injustice issues among those who are “Black, Indigenous, and people of color (BIPOC)” [27].

In a cross-national study using primary survey data collected among neuropsychologists in the United States and Canada [27], the original questionnaire had been divided into separate sections depending on the country of practice [41]. Some differences in the response options for ‘ethnic/racial heritage’ were specified, with respondents in the US asked to self-identify from among seven categories versus those in Canada from among nine categories. Practitioners in the US were asked about “African American/Black” heritage while those in Canada were asked of “African Canadian/Black” heritage. Both survey sections included a response category for “American Indian or Alaskan Native”, whereas the section for Canadian respondents further asked for “First Nations” and “Indigenous Person” identifications. Federal statistical standards in Canada distinguish ethnoracial origins in relation to “visible minority” status, defined as “persons, other than Aboriginal peoples, who are non-Caucasian in race or non-white in colour” [42]. Aboriginal or Indigenous identity is further distinguished as “First Nations (North American Indian), Métis, and Inuk (Inuit)” ancestries [42]. With regard to the study at hand, it was noted that the overall number of survey responses recorded in the Native/Indigenous category (categories) was very small [27].

In Brazil, a study of nursing practitioners based on demographic census data relied on the question asking respondents to self-identify their “colour or race” according to five categories [29]. These categories followed Brazilian federal statistical standards: White (“Branca”), Black (“Preta”), Brown (“Parda”), Yellow/Asian (“Amarela”), or Indigenous (“Indígena”) [43]. Given very low numbers of nurses identified in the latter two categories, the study’s analyses of ethnoracial stratifiers of professional earnings were limited to respondents of “colour or race White, Brown, and Black” [29].

In Norway, in a study using linked administrative registers of educational and labour market characteristics, the data were based on “national groups” according to country of birth and of parents’ birth [22]. In this national context, health labour market equity and integration focused on native-born practitioners versus “non-Western immigrants… born in Asia, Turkey, Eastern Europe, Latin America, and Africa [and] whose parents are also foreign born” [22].

Lastly, in the United Kingdom, 76 ethnicities were captured in the electronic staffing records of the publicly funded healthcare system (i.e., the National Health Service in England), which were grouped when focusing on earnings gaps in terms of White practitioners versus those of “Black, Asian and minority ethnic (BAME)” categorization [39]. In this cultural context, sexual orientation was also included in the health information system as a fundamental diversity element given the health inequalities experienced by lesbian, gay, and bisexual communities compared to heterosexual patients, along with the obligation of public bodies to ensure equality of opportunity for workers [44,45]. The data standard defined sexual orientation in terms of “the stated physical and emotional attraction a person feels towards one sex or another (or both)” [44]. In 2022, employees’ voluntary declaration in their staffing record was considerably higher for ethnic origin than for sexual orientation (98% versus 76%) [45].

### 3.3. Intersecting Impacts on the Gender Wage Gap

Studies in the United States often focused on earnings differences between White and Black practitioners. Two studies documented similar patterns in the contributions of gender and race to physician earnings. Weeks and Wallace found no differences in adjusted annual incomes among family physicians for Black men compared to White men, but significantly lower earnings for White women and even lower for Black women [38]. Frohman and colleagues also found that non-White women surgeons earned significantly less than White women surgeons, alongside similar salaries among non-White and White men surgeons [24]. Meanwhile, Baird and colleagues reported gendered differences in sources of compensation among anesthesiologists in the US, with women more likely to receive salaries and men more often in fee-for-service arrangements [19]; the authors included ethnoracial composition as a control in multivariate models, but did not discuss the variable substantively. Conversely, Hampton and colleagues included gender as a control in their analysis of employed physicians, but did not discuss effects on earnings simultaneously with the role of race; the authors found a small wage premium among racial minorities [26].

The nature of any pay gaps intersecting with gender were found to vary within and across healthcare professions, although the relationships were not necessarily operationalized consistently across studies. For the nursing profession, two studies found a premium in the multivariate wage regressions for Black versus White nurses [21,36], while a third study reported a wage penalty for Black nurses [31], although none presented gender-disaggregated models or included interaction terms between gender and racial group. Self-selection effects by race into female-dominated nursing were ascribed, including the profession having “historically been held in high esteem by Black communities as a viable professional career option for women” [31]. Moore and colleagues assessed that human capital variables (e.g., highest degree, years of experience) and other work characteristics did not produce the same returns in terms of higher earnings for registered nurses from different racial/ethnic groups; decomposition analyses furthered that significant portions of the wage gaps for Black and Hispanic nurses compared with White nurses were unexplained by differences in human capital [30,32].

Among other professions in the US, Gundavarapu and colleagues indicated that ethnoracial diversification of the dentistry workforce was more prominent among women than men [25]. The authors also found a smaller Black/White income disparity among women dentists than among men dentists [25]. Carvajal and colleagues observed differences in the gender composition and earnings of pharmacists across three predominant ethnoracial groups in South Florida, related in part to differences in hours worked and numbers of children in the household [20]. The authors estimated models for earnings determinants separately for each ethnoracial identity, and thus did not attempt to disaggregate gaps between groups [20]. Smith and Jacobson reported that physician assistants are overrepresented by White and women professionals, alongside a persistently large adjusted gender earnings gap demonstrating men’s “structural advantages” even in this female-dominated profession; significant earnings differences were not consistently observed for racial minorities versus White practitioners [34,35].

Klipfel and colleagues used survey data collected among clinical psychologists across North America; results were analyzed and published for US practitioners alone [27]. Gender differences were found in earnings and in income satisfaction, while separate analyses of mean annual earnings did not reveal significant differences across ethnoracial groups. The authors noted some divergence in demographic characteristics and practice patterns of those who “chose not to disclose” their ethnicity/race compared to those who did [27]. In a cross-professional analysis, Frogner and Schwartz also found that the US therapy profession was not paired with a significant wage gap by race/ethnicity, in contrast to the situation observed for nurses, aides, and other healthcare occupations [23]. The authors acknowledged that they did not address selection processes into a given occupation, particularly those such as therapists, which have greater barriers to entry in cost and length of educational requirements [23].

In Brazil, Marinho and colleagues found appreciable racial differences in monthly income, with White nursing professionals and White nurse technicians earning significantly more than their Black and Brown counterparts after adjusting for other factors; gender was included as a control variable in the multivariate analysis but results were not reported distinctly for this female-dominated profession [29].

In Norway, Drange did not discern significant income differences between non-Western immigrant and majority (native-born) physicians and dentists with similar labour market experience, although income trajectories were reported to be “flatter” for women compared to men immigrants [22].

In the United Kingdom, Pudney and Shields modelled nurses’ labour market participation and training histories and found significantly accelerated job promotion trajectories and associated earnings rewards for male over female nurses and for White over Black or Asian nurses [33]. A more recent 2021 study by Woodhams and colleagues of UK physicians explicitly raised the question of how pay gaps by gender and by ethnicity compare “multiplicatively” across overlapping social identities rather than separately, and further considered sexual orientation as salient to differential health labour outcomes [39]. In a multivariate decomposition analysis with White men as the reference category (i.e., usually the highest earning group), lower earnings were observed among Black men that could not be explained by other professional and personal characteristics. Indian men stood as an exception in terms of having higher mean basic pay, a difference that was largely explained by their workforce distribution associated with higher rewards across other personal features—including sexual orientation, nationality, and disability status. Differences in mean age, grade, specialty, and other known factors helped explain most of the pay gaps relative to White men across other gender–ethnicity groups (e.g., Black women, Bangladeshi women). The authors discussed sexual orientation collectively with other personal characteristics, without publication of disaggregated descriptive or inferential statistics, aside from noting the high levels of missing data on sexual orientation in the registry of public sector physicians (34% at the time of the study) [39].

## 4. Discussion

The World Health Organization advocates that health systems should assess pay gaps to ensure fair treatment of health workers and mitigate risks of reduced attractiveness of the health sector to some social groups [16]. While gender-based assessments show nearly universally pay gaps in favour of men, evidence is less clear for many other sociocultural characteristics that may impact opportunities in the health workforce and, ultimately, the provision of services [16]. This scoping review of the peer-reviewed literature sourced 21 original studies on compensation disparities among healthcare practitioners. The studies covered three high immigrant-receiving countries (the United States, the United Kingdom, and Norway, each with ~15% of the population foreign-born) and one country with a lower immigration rate (Brazil, with <1% foreign-born) [46]. Perhaps not surprisingly, the present review found instances of significant earnings differences by gender and other identity characteristics, even in publicly funded health systems. The findings were consistent with previous reviews documenting persistently lower earnings among women healthcare professionals than men (often tens of thousands of dollars less annually), and this across professions and sectors and after adjusting for various human capital and other characteristics [1,2,4].

The results were less uniform regarding earnings differences within and across practitioners’ sociocultural identities. The lack of homogeneous conclusions may relate in part to variations across settings in how race, ethnicity, and ancestry were conceptualized and operationalized. Many studies showed wider wage gaps among women in ethnoracial minority groups than among men. Some suggested that ethnoracial-based selection effects into a given health profession (e.g., Black nurses or Hispanic aides in the United States) could reflect limited career options elsewhere [21,23]. The volume of literature applying multivariate regressions of earnings differences modelled women and men separately, or main ethnoracial groups separately, or incorporated statistical interaction terms for only selected labour market predictors (e.g., union membership)—challenging our understanding of combined effects of gender and social identities. Moreover, the studies tended to group all persons with origins in a large global region (e.g., Africa, South Asia). For example, studies in the US typically collapsed Black practitioners as one “monolithic” group, overlooking diversity within this community (e.g., native-born African Americans and immigrants arriving at different periods and from different parts of Africa, the Caribbean, or elsewhere) [20]. One study included in this review, regarding physician earnings in the United Kingdom, revealed significant differences in how gender interacted with ethnicity upon data disaggregation between those of Indian, Bangladeshi, and Pakistani descent [39]. Our findings on the health workforce echoed a literature review elsewhere highlighting limited evidence focusing on ethnic pay gaps (distinctly from immigrant pay gaps) in national labour markets overall, particularly outside the US [47].

Several studies included in this review purported to consider both gender and ethnocultural identity as salient to earnings differentials, but did not present full sets of regression coefficients or discuss both sets of variables substantively. These results were consistent with reviews elsewhere providing evidence of inadequate reporting of gender-based differences in health and health workforce research, notably large gaps between the mention of sex/gender in studies’ introduction or methods sections versus the reporting of sex-disaggregated data and substantive discussion of the findings and implications [48,49]. Similarly, the sole reviewed study integrating sexual minority status as a potential intersecting characteristic did not disaggregate the statistical results or discuss the direction or magnitude of earnings differences. Overall, few of the studies detailed the completeness of the social identity data used in the analyses, or in turn whether/how the statistical assumption of random distribution of missing data was tested prior to advanced modelling. Only one study explicitly modelled non-disclosure as a separate category of ethnoracial self-identification [27], and one reported a high non-response rate for sexuality as a social identifier [39]. As workplaces become increasingly diverse and attentive to diversity management, more research is needed on less ‘visible’ aspects of workers’ social identities—including sexual orientation, arguably the “last acceptable and remaining prejudice” in modern societies [50].

### Study Strengths and Limitations

As with all scoping reviews [13], this study has some advantages and limitations. While we cast our search strategy widely to provide a transparent mapping of the available research, considering the heterogeneity of healthcare professions, social identity constructs, outcome measures, and analytical designs, it is likely that some studies reflective of the topic were missed in the bibliographic searches. We did not limit our review to a single profession, enabling the inclusion of studies contemplating potential selection effects given the known highly gendered nature of healthcare occupations requiring advanced education. Some occupational specialties were not represented, while others had multiple publications. While our searches capitalized on the variations of terms related to “wage gaps” (see Table A1 in Appendix A), including broader areas of labour outcomes such as “leadership” could have returned more results but was beyond our scope. The studies included in the review leveraged a range of different data sources (e.g., population-based census and surveys, routine staffing and payroll records, ad hoc professional surveys); however, they did not allow us to determine the causes or directionality of gender earnings disparities in relation to other social stratifiers such as ethnicity but also socioeconomic class. Only four countries were covered by the retained studies, one of which was an upper-middle-income country (Brazil) and none were low-income countries. An earlier review of empirical studies analyzing gendered intersections in health and health system outcomes found that the focus in higher-income countries was most often on race/ethnicity, but in lower-income countries, the main focus was often on economic status [51]. We also constrained our review to quantitative research, whereas qualitative methodologies have been found more common in intersectionality articles from low- and middle-income countries [51].

## 5. Conclusions

This scoping study of the peer-reviewed literature confirmed the persistence of a gender wage gap among healthcare practitioners, which cannot be attributed to traditional human capital characteristics or health financing systems. Less clear was how ethnoracial and other social identities may intersect with gender in shaping differentials in health labour market outcomes. As countries worldwide continue to seek solutions for better health workforce equity and inclusion to enhance recruitment and retention, this review highlighted limitations in the evidence base on intersectionality of the gender earnings gap. Our findings suggest that more research is needed: recognizing ethnocultural heterogeneity and disaggregating diversity elements within communities; advancing robust methodologies and transparent reporting in analyses of intersections between gender and social identities on practitioner earnings; and strengthening underlying health workforce information systems to enhance data standards in the capture and use of different social and economic locations of practitioners for more cross-national comparisons, especially in lower-income countries.

## Figures and Tables

**Figure 1 healthcare-13-00273-f001:**
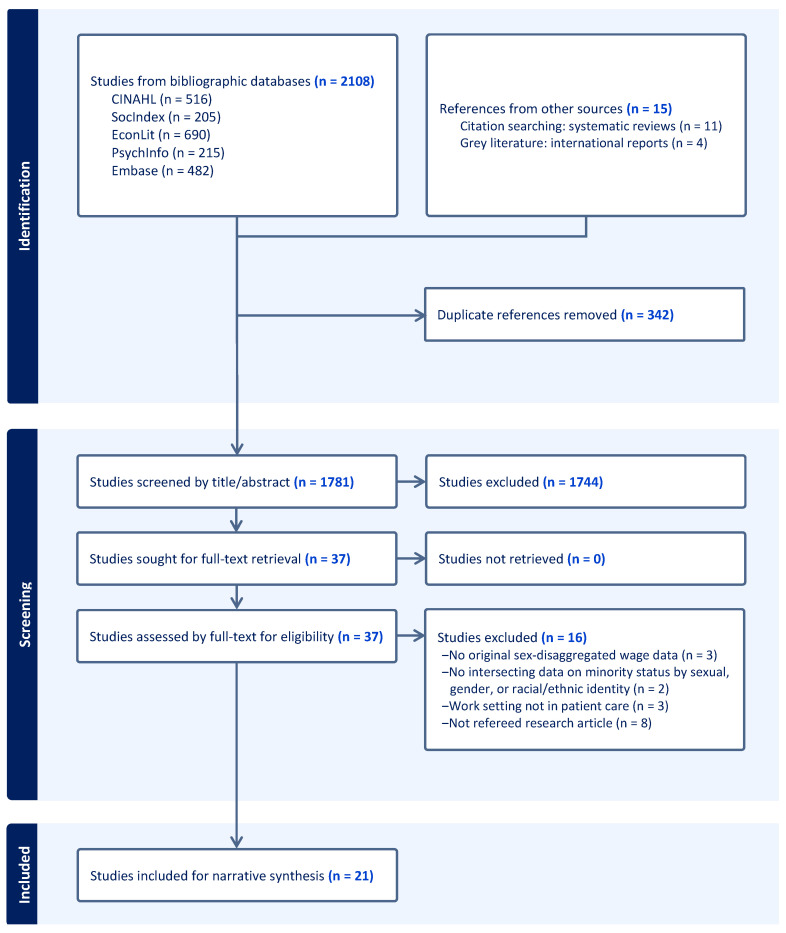
PRISMA flowchart for the selection of articles in the scoping review of intersectionality of the gender wage gap among healthcare professionals.

**Table 1 healthcare-13-00273-t001:** Inclusion and exclusion criteria for the scoping review on intersectionality of the gender wage gap among healthcare professionals.

Criteria	Population	Intervention	Comparisons	Outcome	Study Type
Inclusion	Healthcare professionals in paid clinical roles	Intersecting sociopolitical structures (unmeasured/unmeasurable)	(1) Sexual minority (vs. heterosexual); (2) Gender minority (vs. cisgender); and/or (3) Race, ethnicity, or ancestry	Income/earnings from professional practice	Descriptive, observational, or quasi-experimental design using primary or secondary quantitative research data
Exclusion	Non-clinical professionals; non-direct patient care settings (e.g., academia); unpaid caregivers and volunteers		Men alone or women alone; other sociocultural characteristics alone (e.g., immigration status, geographic region)	Other indicators of productivity, social disadvantage, or differential treatment in the health labour market (perceived or measured)	Theoretical, review, or policy papers without original empirical analysis; non-research papers in refereed journals (e.g., commentaries); student dissertations; other non-refereed sources (e.g., trade journals, professional newsletters, paper deposits)

**Table 2 healthcare-13-00273-t002:** Characteristics of studies included in the scoping review examining intersectionality of the gender wage gap among healthcare professionals.

Study Setting (Author, Year)	Population (Study Sample Size)	Intervention	Comparisons	Outcome Measured	Study Design
Brazil (Marinho et al., 2022) [29]	Nurses and nursing technicians (60,936)	Ethnoracial diversification in opportunities for higher education	3 racial groups: White, Black, Brown	Monthly income	Subsample of nursing occupations from the 2010 Population Census
Norway (Drange, 2013) [22]	Physicians and dentists (8154)	Access and closure in the labour market	2 groups: native-born, non-Western immigrants (from Asia, Turkey, Eastern Europe, Latin America, or Africa)	Earned income and salaries from tax records	Panel data from annual administrative registers, 1993–2002
United Kingdom, England (Pudney and Shields, 2000) [33]	Nurses (8919)	Equality of opportunity and fairness in workplace rewards	3 racial groups: White, Black, Asian	Inferred lifetime earnings by speed to promotion to higher-paying job grades	1994 postal survey of permanent nursing staff in the National Health Service
United Kingdom, England (Woodhams et al., 2021) [39]	Physicians (164,820)	Stereotyping across cross-cutting categories of social difference	7 ethnic groups: White, Black, Indian, Pakistani, Bangladeshi, Chinese/South East Asian, mixed sexual orientation	Inferred pre-tax income by monthly basic pay of job grade	Pooled administrative staffing and payroll records of the National Health Service, 2016–2020
United States (Frogner and Schwartz, 2021) [23]	Physicians; advanced practitioners (dentists, pharmacists, physician assistants); advanced practice registered nurses; registered nurses; licenced nurses; aides/assistants; therapists; technicians; community-based social workers/counsellors) (76,606 across 7 groups)	Structural racism and undervalued labour of diverse ethnoracial groups	5 racial groups: White, Black, Asian/Pacific Islander, Indigenous, multiracial 2 ethnic groups: Hispanic, non-Hispanic	Self-reported annual pre-tax income from wages and salaries	Pooled subsamples of healthcare occupations from the Current Population Survey, 2011–2018
United States (Coomer, 2015) [21]	Registered nurses (159,543)	Historical wage premium for black nurses due to self-selection, experience, shift work, and demand effects	2 racial groups: White, Black	Hourly income [calculated as annual income ÷ (hours*weeks)]	Pooled cross-sections from the National Sample Survey of Registered Nurses, 1984–2008
United States (McGregory, 2013) [31]	Registered nurses (20,842)	Real or perceived racial inequalities in the labour market	2 racial groups: White, Black	Weekly earnings	Pooled subsamples of nurses employed in healthcare facilities from the Current Population Survey, 1994–2006
United States, New York City (McGinnis and Moore, 2009) [30]	Registered nurses (2690)	Developing a more culturally competent health workforce	4 ethnoracial groups: non-Hispanic White, Black/African American, Hispanic/Latino, Asian/Pacific Islander	Annual gross salary, excluding overtime (12 range categories)	Survey of registered nurses employed in hospitals by the Center for Health Workforce Studies, 2006–2007
United States Moore and Continelli, 2016) [32]	Registered nurses (4028)	Improving diversity and cultural competence in healthcare	4 ethnoracial groups: White, Black, Hispanic, Asian	Hourly wages	Subsample of hospital nurses in metropolitan areas from the 2008 National Sample Survey of Registered Nurses
United States (Wagner et al., 2021) [36]	Registered nurses (15,373)	Diversification of the older population and cultural competency in long-term care	5 ethnoracial groups: White, Black, Hispanic, Asian, other	Hourly wage in the principal nursing position [calculated as total annual earnings ÷ total annual hours]	Subsample of registered nurses in long-term care from the 2018 National Sample Survey of Registered Nurses
United States (Hampton and Heywood, 1999) [26]	Physicians (1872)	Job satisfaction and relative deprivation	2 groups: non-Hispanic White, racial minority	Employment earnings	1987 survey of early career employee physicians by the American Medical Association
United States (Weeks and Wallace, 2006) [38]	Physicians (977)	Practice arrangements and likelihood to care for underserved groups	2 racial groups: White, Black	Annual net income	Pooled samples from annual telephone surveys of family physicians by the American Medical Association, 1992–2001
United States (Weeks et al., 2009) [37]	Physicians (1179)	Practice arrangements and likelihood to care for underserved groups	4 racial groups: White/Caucasian, Black/African American, Hispanic, Asian or Pacific Islander	Annual net income	Panel data of primary care physicians from the Community Tracking Study Physician Surveys, 1998–1999, 2001–2002 and 2004–2005
United States (Kornrich, 2009) [28]	Physicians (4089)	Historical racially embedded work processes and patient–provider racial matching	2 racial groups: non-Hispanic White, non-Hispanic Black	Weekly earnings	Practice Patterns of Young Physicians Survey by the American Medical Association, 1991
United States (Frohman et al., 2015) [24]	Surgeons (194)	Influences on medical students’ choice of specialty	2 groups: White, non-White (Asian, African American, Hispanic, and other)	Self-reported annual income and perceptions of salary	Online survey distributed to participants at selected surgical conferences, 2011–2012
United States (Baird et al., 2015) [19]	Anesthesiologists (6783)	Compensation structures and inflexible practice scheduling	5 racial groups: White, Hispanic, Black, Asian, other	Compensation from fee-for-service, salary and bonus (dollars per hour)	Repeated cross-sectional national surveys of anesthesia practitioners, 2007 and 2013
United States (Smith and Jacobson, 2016) [35]	Physician assistants (15,102)	Historical racial discrimination	5 ethnoracial groups: non-Hispanic White, non-Hispanic Black, Hispanic, Asian, other	Annual income from the primary employer	2009 survey by the American Academy of Physician Assistants
United States (Smith and Jacobson, 2018) [34]	Physician assistants (3642)	Historical racism and discrimination	5 ethnoracial groups: non-Hispanic White, non-Hispanic Black, Hispanic, Asian, other	Personal annual income	Pooled subsample of physician assistants in the American Community Survey, 2010–2012
United States (Gundavarapu et al., 2023) [25]	Dentists (143,671)	Gender roles, job flexibility, and workforce continuity	5 racial groups: White, Black, Asian, Hispanic, other	Personal annual earned income	Pooled subsample of dentists in the American Community Survey, 2014–2018
United States, South Florida (Carvajal et al., 2013) [20]	Pharmacists (1139)	Ethnic diversification of the population	3 ethnic groups: non-Hispanic White, Black, Hispanic	Self-reported earnings from wages and salaries	2006 postal survey of licenced pharmacists
United States and Canada (Klipfel et al., 2023) [27]	Neuropsychologists (1677)	Pandemic-related job disruptions and increased attention to social justice issues	7 ethnoracial groups: White, Black/African American, American Indian or Alaskan Native, Asian or Pacific Islander, Hispanic/Latino(a), biracial/multiethnic/multiracial, or chose not to disclose	Self-reported annual income and hourly fees charged	2020 online survey distributed to clinical neuropsychologists and postdoctoral trainees via North American professional organization membership lists

Source: Adapted from [19,20,21,22,23,24,25,26,27,28,29,30,31,32,33,34,35,36,37,38,39].

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
