# Peer review of "Intersectionality of the Gender Wage Gap Among Healthcare Professionals: A Scoping Review"

_healthcare, 2025, doi:10.3390/healthcare13030273_

Round 1
Reviewer 1 Report
Comments and Suggestions for Authors
We appreciate that the paper approaches a subject that has been insufficiently explored in the literature and is an essential issue from an ethical point of view.
The Abstract is well-structured, and the scoping review respects the protocol/methodology.
Despite these strengths, we suggest that the authors fix the following issues to improve the quality of the paper:
- It would be appropriate to narrow the range of healthcare professionals in this analysis (as it was researched, the categories included are too diverse, it is a too large approach, and there are not enough studies for each category);
- There is no uniformity in using the key terms - e.g., earnings might differ from wages;
- The authors should consider that the analyzed countries have different socioeconomic and cultural specificities. There are insufficient arguments as to why the research focuses on these countries. Moreover, the wage level is influenced by many things, including years of experience in the field;
- In the Discussion section, limit the explanations to discussing the results in light of other research/studies (similarity, differences, etc.). Highlight the study's limitations in a separate section;
- Add the structure of the paper (the main parts of it) at the end of the introduction section;
- Please check the journal's recommendations regarding citing references in the text; these recommendations are not well-addressed;
- The conclusions should be more consistent.
Author Response
COMMENT 1: We appreciate that the paper approaches a subject that has been insufficiently explored in the literature and is an essential issue from an ethical point of view. The Abstract is well-structured, and the scoping review respects the protocol/methodology. Despite these strengths, we suggest that the authors fix the following issues to improve the quality of the paper: - It would be appropriate to narrow the range of healthcare professionals in this analysis (as it was researched, the categories included are too diverse, it is a too large approach, and there are not enough studies for each category);
RESPONSE 1: We thank the Reviewer for the thoughtful consideration of our manuscript. We have made a number of revisions, as indicated in red font. We have better specified in the Discussion that “We did not limit our review to a single profession, enabling inclusion of studies contemplating potential selection effects given the known highly gendered nature of healthcare occupations requiring advanced education” (lines 389-391). We appreciate the acknowledgement that there were not enough studies in each category of health workers, but please note we systematically searched the literature using a range of keywords (e.g. per Appendix Table A1 documenting our search strategy) – as such, we do not believe we would have found any more studies for a particular cadre had we been more restrictive. We have documented which study covered which profession(s) in Table 2 and its associated interpretations. As better elaborated in the Introduction, we have also noted the importance of considering differences in professional trajectories across a range of occupations including “traditionally male-dominated medicine, dentistry, and pharmacy versus female-dominated nursing and midwifery” (lines 45-47). Moreover, addressing gender gaps is an issue raised by many different health professional associations (e.g. lines 47-53).
COMMENT 2: There is no uniformity in using the key terms - e.g., earnings might differ from wages;
RESPONSE 2: We have better clarified in the Methods that “Although health worker remuneration can comprise different types of payments, we followed the common practice of using the terms earnings, incomes, wages, and salaries interchangeably” (lines 119-121). As noted in Appendix Table A1, we searched using a variety of terms as itself a reflection of the literature.
COMMENT 3: The authors should consider that the analyzed countries have different socioeconomic and cultural specificities. There are insufficient arguments as to why the research focuses on these countries. Moreover, the wage level is influenced by many things, including years of experience in the field;
RESPONSE 3: The included studies reflected the results of our scoping review, not an intentional choice, as when conducting our searches “No restrictions were placed by world region, country income level, healthcare financing system” (lines 117-118). We have separated in the Discussion a distinct subsection on ‘Study strengths and limitations’, where we explicitly acknowledge: “Only four countries were covered by the retained studies, one of which was an upper-middle income country (Brazil) and none were low-income countries” (lines 400-402). We have also better placed this limitation within the context of the wider literature, including that: (1) “Our findings on the health workforce echoed a literature review elsewhere highlighting limited evidence focusing on ethnic pay gaps (distinctly from immigrant pay gaps) in national labour markets overall, particularly outside the US” (lines 361-364, supported by a new reference), and (2) we focused “our review to quantitative research, whereas qualitative methodologies have been found more common in intersectionality articles from low- and middle-income countries” (lines 405-407, backed by appropriate referencing).
COMMENT 4: In the Discussion section, limit the explanations to discussing the results in light of other research/studies (similarity, differences, etc.). Highlight the study's limitations in a separate section;
RESPONSE 4: As noted above, we have separated in the Discussion a distinct subsection on ‘Study strengths and limitations’. We have also better limited the explanations of our results in relation to other similar types of review studies (in some cases supported by new references).
COMMENT 5: Add the structure of the paper (the main parts of it) at the end of the introduction section;
RESPONSE 5: The structure of the paper is identified/justified at the end of the Introduction section ( lines 90-94, supported by appropriate referencing): “Following this introduction, the review is structured according to the framework for scoping studies by Arksey and O’Malley (13): identification of the research question; identification and selection of relevant studies; charting the data; and collating, summarizing, and reporting the results.”
COMMENT 6: Please check the journal's recommendations regarding citing references in the text; these recommendations are not well-addressed;
RESPONSE 6: We have re-reviewed the recommendations for referencing on the Healthcare MDPI Instructions for Authors, which indicate: “References must be numbered in order of appearance in the text (including table captions and figure legends) and listed individually at the end of the manuscript. … you may use reference style 9. below.” We used a bibliography software package (Zotero) to follow the numbering style and avoid duplicated references, and are uncertain where the recommendations were not followed.
COMMENT 7: The conclusions should be more consistent.
RESPONSE 7: We have better prefaced the Conclusions with a synthesis of our key findings, notably “This scoping study of the peer-reviewed literature confirmed the persistence of a gender wage gap among healthcare practitioners, which cannot be attributed to traditional human capital characteristics or health financing systems. Less clear was how ethnoracial and other social identities may intersect with gender in shaping differentials in health labour market outcomes” (lines 410-414).
Reviewer 2 Report
Comments and Suggestions for Authors
This is a useful scoping review of studies about how gender wage gaps and wage gaps between ethnic groups intersect in the health care sector. The selection of papers is well documented and seems valid. However, there are some omissions which should be covered, or should be justified. They are listed below in order of importance.
1) Some studies are never referred to in the Results section (outside Table 2): 26, 30, 32, 33, 34, 35, 36. It is possible that they add nothing to the other papers, but then this should be explicitly stated.
2) An important distinction in studies of the gender wage gap is between the raw wage gap and the 'net' wage gap when controlling for observable characteristics, e.g. education, experience, working time, etc. Table 2 and the discussion do not mention to what extent the wage gaps identified in the studies were net of other factors.
3) Intersectionality is rather vague concept, which can cover several mechanisms, e.g. ethnic wage gaps as an explanatory factor of the gender wage gap, the gender wage gap as an explanatory factor of ethnic wage gap, the gender wage gap occurring only among particular ethnic groups (or particular professions). There may be others. The discussion seems to favour an interaction approach. A brief discussion about what intersectionality can mean would be useful.
4) The conclusions recommends that "health workforce information systems [should] capture different social and economic locations of practitioners more fluidly and broadly," This may run up against privacy considerations. Many people may feel that there is no need at all for their employers to know about their sexual orientation, and some may have the same feeling about their ethnic background. Also, I do not know what you mean by "fluidly" in this context, a word also used earlier in the text.
Author Response
COMMENT 1: This is a useful scoping review of studies about how gender wage gaps and wage gaps between ethnic groups intersect in the health care sector. The selection of papers is well documented and seems valid. However, there are some omissions which should be covered, or should be justified. 1) Some studies are never referred to in the Results section (outside Table 2): 26, 30, 32, 33, 34, 35, 36. It is possible that they add nothing to the other papers, but then this should be explicitly stated.
RESPONSE 1: We thank the Reviewer for the thoughtful consideration of our manuscript. We have made a number of revisions, as indicated in red font. We have corrected the oversight (with apologies) by explicitly discussing in the Results the previously-overlooked references: #26 (lines 256-259), #30 & 32 (lines 268-273), #33 (lines 308-311), # 34 & 35 (lines 282-286), and #36 (lines 262-265).
COMMENT 2) An important distinction in studies of the gender wage gap is between the raw wage gap and the 'net' wage gap when controlling for observable characteristics, e.g. education, experience, working time, etc. Table 2 and the discussion do not mention to what extent the wage gaps identified in the studies were net of other factors.
RESPONSE 2. We have added in the Results more details on the types of characteristics included in the studies, for example “Moore and colleagues assessed that human capital variables (e.g., highest degree, years of experience) and other work characteristics did not produce the same returns in terms of higher earnings for registered nurses from different racial/ethnic groups” (lines 268-271). We did not intend to quantify the extent of raw versus net wage gaps, and have better specified in the Methods (backed by appropriate referencing) that we used a narrative synthesis approach: “Because of the heterogeneity typical in pay gap research of target populations, outcome measurements, confounding factors examined, and research protocols and norms (2), we did not expect to be able to perform a meta-analysis” (lines 151-154).
COMMENT 3) Intersectionality is rather vague concept, which can cover several mechanisms, e.g. ethnic wage gaps as an explanatory factor of the gender wage gap, the gender wage gap as an explanatory factor of ethnic wage gap, the gender wage gap occurring only among particular ethnic groups (or particular professions). There may be others. The discussion seems to favour an interaction approach. A brief discussion about what intersectionality can mean would be useful.
RESPONSE 3. We have added the following in the Methods section: “The WHO acknowledges a broad range of societal drivers may intersect with gender in influencing and reinforcing discrimination and inequities in health labour markets – including race/ethnicity and sexual orientation but also class and other stratifiers of access to opportunities, remuneration, and power in decision-making” (lines 126-130). We have also specified that we considered social identities as those “often compared in relation to a dominant group in a culture” (lines 145-146), and that we included any type of “quantitative” studies using “statistical interaction or decomposition methods on population-level microdata” (lines 147-150). We have reframed the Discussion, including removing any suggestion that we focused on any given statistical method. We also explicitly identified as a study limitation that our results “did not allow us to determine the causes or directionality of gender earnings disparities in relation to other social stratifiers of gender differences such as ethnicity but also socioeconomic class” (lines 398-400).
COMMENT 4) The conclusions recommends that "health workforce information systems [should] capture different social and economic locations of practitioners more fluidly and broadly," This may run up against privacy considerations. Many people may feel that there is no need at all for their employers to know about their sexual orientation, and some may have the same feeling about their ethnic background. Also, I do not know what you mean by "fluidly" in this context, a word also used earlier in the text.
RESPONSE 4. We have reframed the Conclusions with a recommendation for “strengthening underlying health workforce information systems to enhance data standards in the capture and use of different social and economic locations of practitioners for more cross-national comparisons, especially in lower income countries” (lines 421-423). We have added in the Discussion (as backed by a new reference): “As workplaces become increasingly diverse and attentive to diversity management, more research is needed on less ‘visible’ aspects of workers’ social identities – including sexual orientation, arguably the “last acceptable and remaining prejudice” in modern societies” (lines 379-382). We have also added upfront in the Introduction (referenced appropriately): “Disentangling the drivers of differential treatment in the health workforce is challenging, although it has been argued the more employers capture information about their workers, the less likely salaries may be dependent on observable personal characteristics” (lines 80-83).
Reviewer 3 Report
Comments and Suggestions for Authors
This paper provides a review of the wage gap between women and men in the healthcare workforce.
1. The gender wage gap in the labor market has been the focus of economists. Why did the authors choose gender wages in the healthcare workforce as the focus of this study? What are the different characteristics of the gender wage gap in this industry? The introduction of this paper does not clearly address the above questions.
2. Five databases, CINAHL, EconLit, SocIndex, and PsychInfo, were searched for relevant literature. Why didn't you search for relevant literature in more databases such as Google Scholar, Web of Science, etc.?
3. In lines 97-99, the paper mentions that the keywords used in the literature search process are too few and should include “wage gap,” “gender gap,” and so on;
4. From the third part of the text, this paper is only a cursory review of a small amount of literature, and the process lacks sufficient detail and depth.
5. The gender wage gap in the medical industry is closely related to the model of the healthcare system. But 81% of the literature used in the article is from the United States. The conclusion of this literature review may be biased.
6. This paper does not get the conclusion of academic value.
Author Response
COMMENT 1. This paper provides a review of the wage gap between women and men in the healthcare workforce. The gender wage gap in the labor market has been the focus of economists. Why did the authors choose gender wages in the healthcare workforce as the focus of this study? What are the different characteristics of the gender wage gap in this industry? The introduction of this paper does not clearly address the above questions.
RESPONSE 1. We thank the Reviewer for the thoughtful consideration of our manuscript. We have made a number of revisions, as indicated in red font. We have better specified upfront the wider interest in the gender wage gap in the healthcare workforce: “Given that women account for the strong majority (over two-thirds) of the health and social workforce worldwide, such underinvestment in health practitioners undermines the prospects of achieving the Sustainable Development Goals and universal health coverage” (lines 40-43). We have also added in the Introduction: “Many workplace differentials in economic rewards observed across sectors (public, private, and multi-sector environments) and around the globe cannot be explained by human capital and skill competencies alone, but rather to effects of non-work-related factors which may induce pay gaps, such as gender and social discrimination (2). Unequal pay for work of comparable value can lead to health worker turnover and burnout, negatively impacting quality of care and health system performance (1)” (lines 60-65).
We have also revamped the Discussion, and better contextualized our study: “The World Health Organization advocates that health systems should assess pay gaps to ensure fair treatment of health workers and mitigate risks of reduced attractiveness of the health sector to some social groups (16). While gender-based assessments show nearly universally pay gaps in favour of men, evidence is less clear for many other sociocultural characteristics that may impact opportunities in the health workforce and, ultimately, the provision of services” (lines 328-333).
COMMENT 2. Five databases, CINAHL, EconLit, SocIndex, and PsychInfo, were searched for relevant literature. Why didn't you search for relevant literature in more databases such as Google Scholar, Web of Science, etc.?
RESPONSE 2. We have better specified in the Methods that “We systematically searched for relevant peer-reviewed publications in five specialized (biomedical and non-biomedical) abstract and citation databases: Embase, CINAHL, EconLit, SocIndex, and PsychInfo. The search strategy was developed with the advice and assistance of a library professional.” The library professionals at our university tend to advise that multidisciplinary search engines (e.g. Google Scholar) may be useful for initial searches, but are less amenable to structured searches with specific inclusion/exclusion criteria characteristic of scoping reviews such as ours.
COMMENT 3. In lines 97-99, the paper mentions that the keywords used in the literature search process are too few and should include “wage gap,” “gender gap,” and so on;
RESPONSE 3. We note in the Methods that we used these terms “and related nomenclature” (lines 133-114) and that “A complete illustrative search strategy for one of the databases can be found in the Appendix (Table A.1)”. We have additionally prefaced this with: “Although health worker remuneration can comprise different types of payments, we followed the common practice of using the terms earnings, incomes, wages, and salaries interchangeably” (lines 119-122). We have also better specified in the Discussion that “our searches capitalized on variations of terms related to “wage gaps” (see Table A.1 in Appendix)” (lines 393-394).
COMMENT 4. From the third part of the text, this paper is only a cursory review of a small amount of literature, and the process lacks sufficient detail and depth.
RESPONSE 4. We have added more details in the Results section 3.3 (please see above for our response to comments from Reviewer 2). We have also added in the Discussion that the small volume of studies found in our review “echoed a literature review elsewhere highlighting limited evidence focusing on ethnic pay gaps (distinctly from immigrant pay gaps) in national labour markets overall, particularly outside the US” (lines 361-364, backed by a new reference), and that “more research is needed on less ‘visible’ aspects of workers’ social identities – including sexual orientation” (lines 380-382, backed by a new reference).
COMMENT 5. The gender wage gap in the medical industry is closely related to the model of the healthcare system. But 81% of the literature used in the article is from the United States. The conclusion of this literature review may be biased.
RESPONSE 5. As noted in the point above, our findings “echoed a literature review elsewhere highlighting limited evidence focusing on ethnic pay gaps… particularly outside the US” (lines 361-364). We have added a ‘Study strengths and limitations’ subsection in the Discussion, where we now explicitly acknowledge: “Only four countries were covered by the retained studies, one of which was an upper-middle income country (Brazil) and none were low-income countries. An earlier review of empirical studies analyzing gendered intersections in health and health system outcomes found the focus in higher income countries was most often on race/ethnicity, but in lower income countries the main focus was often on economic status (51). We also constrained our review to quantitative research, whereas qualitative methodologies have been found more common in intersectionality articles from low- and middle-income countries (51)” (lines 400-407).
COMMENT 6. This paper does not get the conclusion of academic value.
RESPONSE 6. We have a better prefaced the Conclusions (please see above for our response to comments from Reviewer 1), notably by adding an overarching summary of our review findings: “This scoping study of the peer-reviewed literature confirmed the persistence of a gender wage gap among healthcare practitioners, which cannot be attributed to traditional human capital characteristics or health financing systems. Less clear was how ethnoracial and other social identities may intersect with gender in shaping differentials in health labour market outcomes” (lines 410-414).
Round 2
Reviewer 1 Report
Comments and Suggestions for Authors
We appreciate the authors' efforts in addressing the reviewer's recommendations, clarifying the highlighted aspects, and improving the overall quality of the paper. We additionally recommend that the authors add the source below the tables, even if it is their elaboration.
Author Response
COMMENT FROM REVIEWER 1: We appreciate the authors' efforts in addressing the reviewer's recommendations, clarifying the highlighted aspects, and improving the overall quality of the paper. We additionally recommend that the authors add the source below the tables, even if it is their elaboration.
RESPONSE 1: We thank the Reviewer for the ongoing attention to our submission. We double-checked the PRISMA reporting guidelines as well as a few of the most recent review studies published in Healthcare, and do not see either guidelines or examples for adding the sources below the Tables characterising the synthesized studies, already specified in the table contents and in the test. Nevertheless, we have added a footnote to Table 2 with the references, and trust the Healthcare editors to correct if there are any contraventions to the journal's house style. Please note there is no source attributable to either Table 1 or Figure 1, as these are exclusively the authors' original work.
Reviewer 2 Report
Comments and Suggestions for Authors
My comments have been adequately addressed. I have only one minor remark. I found the statement "the more employers capture information about their workers, the less likely salaries may be dependent on observable personal characteristics" puzzling. It is unobjectionable if salaries depend on characteristics such as education, experience and place of work. So please qualify or clarify.
Author Response
COMMENT 1: My comments have been adequately addressed. I have only one minor remark. I found the statement "the more employers capture information about their workers, the less likely salaries may be dependent on observable personal characteristics" puzzling. It is unobjectionable if salaries depend on characteristics such as education, experience and place of work. So please qualify or clarify.
RESPONSE 1: We thank the Reviewer for the ongoing consideration of our submission. We have clarified the statement (lines 80-85, based on a reference from the literature) as follows: "Despite widespread evidence of differential treatment in the health workforce, disentangling the drivers of earnings gaps against what would be predicted statistically in truly ‘gender-blind’ and ‘colour-blind’ societies is challenging; however, it has been argued the more employers capture information about their workers, the more likely salaries may be dependent on measures of productivity and less so on readily observable personal characteristics (10)."
Reviewer 3 Report
Comments and Suggestions for Authors
The quality of the paper has been significantly improved after revision.
Author Response
COMMENT 1: The quality of the paper has been significantly improved after revision.
RESPONSE 1: We thank the Reviewer for the ongoing consideration and appreciation of our submission.